# TRPM8: A Therapeutic Target for Neuroinflammatory Symptoms Induced by Severe Dry Eye Disease

**DOI:** 10.3390/ijms21228756

**Published:** 2020-11-19

**Authors:** Darine Fakih, Christophe Baudouin, Annabelle Réaux-Le Goazigo, Stéphane Mélik Parsadaniantz

**Affiliations:** 1Sorbonne Université, INSERM, CNRS, Institut de la Vision, 17 rue Moreau, F-75012 Paris, France; darine.fakih@inserm.fr (D.F.); cbaudouin@15-20.fr (C.B.); annabelle.reaux@inserm.fr (A.R.-L.G.); 2R&D Department, Laboratoires Théa, 12 rue Louis Biérot, F-63000 Clermont-Ferrand, France; 3CHNO des Quinze-Vingts, INSERM-DGOS CIC 1423, 17 rue Moreau, F-75012 Paris, France; 4Department of Ophthalmology, Ambroise Paré Hospital, AP-HP, University of Versailles Saint-Quentin-en-Yvelines, 9 avenue Charles de Gaulle, F-92100 Boulogne-Billancourt, France

**Keywords:** cornea, electrophysiology, inflammation, dry eye, TRPM8 antagonist

## Abstract

Dry eye disease (DED) is commonly associated with ocular surface inflammation and pain. In this study, we evaluated the effectiveness of repeated instillations of transient receptor potential melastatin 8 (TRPM8) ion channel antagonist M8-B on a mouse model of severe DED induced by the excision of extra-orbital lacrimal and Harderian glands. M8-B was topically administered twice a day from day 7 until day 21 after surgery. Cold and mechanical corneal sensitivities and spontaneous ocular pain were monitored at day 21. Ongoing and cold-evoked ciliary nerve activities were next evaluated by electrophysiological multi-unit extracellular recording. Corneal inflammation and expression of genes related to neuropathic pain and inflammation were assessed in the trigeminal ganglion. We found that DED mice developed a cold allodynia consistent with higher TRPM8 mRNA expression in the trigeminal ganglion (TG). Chronic M8-B instillations markedly reversed both the corneal mechanical allodynia and spontaneous ocular pain commonly associated with persistent DED. M8-B instillations also diminished the sustained spontaneous and cold-evoked ciliary nerve activities observed in DED mice as well as inflammation in the cornea and TG. Overall, our study provides new insight into the effectiveness of TRPM8 blockade for alleviating corneal pain syndrome associated with severe DED, opening a new avenue for ocular pain management.

## 1. Introduction

Dry eye disease (DED) is a chronic ocular disease characterized by dryness, discomfort, burning sensation, and pain in the eye [1]. The prevalence of DED can reach up to 50% in people aged 65 years and over [2]. DED occurs due to deficient tear production or excessive evaporation, causing damage to the ocular surface. Persistent ocular surface pain, which can occur in moderate to severe DED, has gained recognition due to its increasing prevalence, morbidity, and the resulting social burden [3]. However, the underlying mechanisms are not fully understood and, to date, no therapeutic strategy is satisfactory for alleviating ocular pain associated with chronic DED.

The cornea is a transparent avascular tissue considered to have the densest nociceptive nerve terminals of all body tissues [3,4,5]. The nociceptive innervation of the cornea is provided by the ciliary nerve, where neurons are located in the ophthalmic branch of the trigeminal ganglion (TG) [3,6,7]. Different types of receptors (mechano-nociceptors, polymodal nociceptors, and thermoreceptors) coexist on a single corneal nerve ending [8]. Corneal cold thermoreceptors, which represent 50% of the total population of corneal sensory neurons in mice, detect ocular dryness and maintain tear homeostasis by adjusting the basal tearing rate and spontaneous blinking frequency [9,10,11,12]. The transient receptor potential melastatin 8 (TRPM8) receptor is a cold- and menthol-sensitive non-selective cation channel [13] that is expressed by a subpopulation of cold sensory trigeminal neurons [14,15,16]. Genetic deletion of TRPM8 revealed that TRPM8−/− mice are deficient in unpleasant cold sensitivity, confirming the implication of this channel in cold-pain sensation [17,18,19]. Cold allodynia is a common feature of neuropathic pain [20,21] and numerous preclinical studies demonstrated the essential function of TRPM8 in neuropathic pain perception [22,23,24,25,26]. In this regard, pharmacological studies have demonstrated that TRPM8 ligands or inhibitors/blockers attenuate pain sensation in numerous somatic pain models. TRPM8 antagonists were shown to efficiently alleviate acute and chronic pain [27,28,29,30], whereas the TRPM8 agonist may present significant antiallodynic activity through an excessive activation of TRPM8, leading to its downregulation [31].

In DED, noxious (thermal, mechanical, and chemical) stimuli and inflammation enhance the activation of corneal sensory neurons terminals, leading to the intensification of pain sensations [8]. Electrophysiological recording studies showed that DED increases ongoing corneal activities, cold nociceptors responsiveness, and TG cold neurons sensitivity in models of DED in guinea pigs and rats [32,33]. Acute ocular instillation of TRPM8 antagonist was reported to decrease the cold sensitivity of corneal neurons to dry stimulations in wild-type rat [34]. TRPM8 plays an essential role in the mechanisms of tearing and blinking [10,35,36]. Some studies supported the hypothesis that TRPM8 agonist may alleviate chronic pain and dryness in DED [37], whereas others reported that TRPM8 over-activation could result in the desensitization of trigeminal neurons, leading to reductions in tear production, which could have a more deleterious effect on DED patients [33]. Preclinical and clinical studies have reported that ocular menthol application at low doses produces a cooling sensation, whereas at higher doses, it evokes burning, irritation, and pain [35,38,39]. Topical application of menthol (0.2 mM) to human blunted cold-induced perceptions while the irritation sensation caused by low temperature stimuli endured [40].

The aim of this study was to provide new insight into the effectiveness of TRPM8 antagonist topical treatment on corneal neurosensory abnormality syndrome induced by severe DED. Herein, we report higher TRPM8 mRNA expression in the ophthalmic branch of TG and corneal cold hypersensitivity in a mouse model of severe DED [41]. We found in an ex vivo eye preparation of DED mice that single application of TRPM8 antagonist (M8-B) at 20 and 30 µM decreased cold-evoked ciliary nerve activity compared to a saline-treated DED group. We further found in vivo that chronic topical instillations of (20 µM) M8-B twice daily for 15 days in DED mice decreased (1) the spontaneous and cold-evoked activity of corneal nerve in ex vivo eyes preparations, (2) corneal inflammation, (3) expression of genes related to neuropathic and inflammatory pain in the TG, and (4) corneal mechanical allodynia and spontaneous ocular pain. The research design of this study is presented in Figure 1.

## 2. Results

### 2.1. Severe DED Induced Corneal Cold Hypersensitivity Correlated with an Increase in TRPM8 mRNA Expression in the TG

We first investigated whether a severe DED induced cold hypersensitivity at day 21. To do so, a drop of 50 µM menthol, a TRPM8 agonist, was directly instilled to the cornea and the eye closing ratio was then monitored. Figure 2A shows the eye closing ratio decreased by 23% in DED mice compared to sham mice (0.52 ± 0.03 vs. 0.39 ± 0.03, *p* < 0.05), suggesting that DED animals develop cold hypersensitivity. We next investigated whether this sensory abnormality could be due to changes in TRPM8 mRNA expression in the TG from DED mice using RNA scope in situ hybridization. Semi-quantitative analysis of the area covered by the TRPM8 mRNA fluorescent probe showed that TRPM8 mRNA expression was significantly higher (+254.84%) in the ophthalmic branch of the TG of DED mice compared to sham animals (sham: 0.31 ± 0.03 vs. DED: 1.10 ± 0.21, *p* < 0.05; Figure 2B).

### 2.2. Instillation of M8-B Reduced Ongoing and Cold-Induced Electrical Activities of Thermoreceptors in Ex Vivo Eye Preparations from DED Animals

Ongoing and evoked electrical activities of the ciliary nerve (Figure 3A) were recorded at day 21 post-surgery using a multi-unit extracellular recording in ex vivo eye preparations from sham and DED animals. First, we analyzed the spontaneous nerve activities while perfusing the cornea with superfusion saline solution at a temperature set to 32–34 °C. Figure 3B shows histograms representing the number of impulses (imp) per second (sec) from sham and DED mice at 32–34 °C. DED mice exhibited a higher spontaneous activity (+72.29%) compared to sham animals (sham + vehicle: 55.25 ± 4.34 imp/sec, DED + vehicle: 95.19 ± 7.38 imp/sec, *p* < 0.00001; Figure 3B).

Next, we recorded the spontaneous activity of the ciliary nerve while perfusing the cornea with saline solution at 32–34 °C containing 10, 20, or 30 µM of M8-B. The spontaneous activity of the ciliary nerve remained unchanged for all M8-B concentrations tested at 32 °C (sham + vehicle: 55.25 ± 4.34 imp/sec vs. sham + M8-B (10 µM) 68.60 ± 10.09 imp/sec vs. sham + M8-B (20 µM) 68.40 ± 17.98 imp/sec vs. sham + M8-B (30 µM) 72.00 ± 14.45 imp/sec, *p* > 0.05; Figure 3A) and DED + vehicle (95.19 ± 7.38 imp/sec) vs. DED + M8-B (10 µM; 88.80 ± 12.18 imp/sec) vs. DED + M8-B (20 µM; 74.80 ± 19.00 imp/sec) vs. DED + M8-B (30 µM; 76.20 ± 20.27 imp/sec; *p* > 0.05; Figure 3B).

Then, we recorded the cold-evoked activities in response to a cold stimulation of the ciliary nerve while perfusing the cornea with superfusion saline solution with temperature set to 20 °C. Figure 3C shows histograms representing the number of imp per sec from sham and DED mice. At 20 °C, we observed a higher nerve electrical activity (+36.88%) in DED ex vivo eye preparations compared to sham animals (sham + vehicle 95.19 ± 7.38 imp/sec vs. DED + vehicle 130.30 ± 8.66 imp/sec, *p* < 0.001).

The acute application of M8-B at 10, 20, and 30 µM on ex vivo eye preparations from sham animals did not significantly alter the cold-evoked (at 20 °C) ciliary nerve activities (sham + vehicle 95.19 ± 7.38 imp/sec vs. sham + M8-B (10 µM) 88.80 ± 12.18 imp/sec vs. sham + M8-B (20 µM) 74.80 ± 19.00 imp/sec vs. sham + M8-B (30 µM) 76.20 ± 20.27 imp/sec, *p* > 0.05; Figure 3C). In contrast, the acute treatment of the ex vivo eye preparation from DED mice with M8-B at 20 or 30 µM in superfusion medium set to 20 °C significantly blocked the cold-induced activation of nociceptors by 44.05% and 40.42%, respectively (DED + vehicle 130.30 ± 8.66 imp/sec vs. DED + M8-B (10 µM) 88.12 ± 16.13 imp/sec, *p* > 0.05; vs. DED + M8-B (20 µM) 72.90 ± 16.34 imp/sec, *p* < 0.01; vs. DED + M8-B (30 µM) 77.63 ± 15.73 imp/sec, *p* < 0.05; Figure 3C). We found no statistical difference in the cold (20 °C)-evoked activities of the ciliary nerve between sham and DED groups while treating the cornea with M8-B at 10, 20, and 30 µM (Figure 3C).

### 2.3. Chronic Topical Treatment with M8-B Reduced Corneal Cold Nociceptor Sensitization in DED Animals

To decrease cold-sensitization of DED mice after an acute application to the ex vivo eye preparation, M8-B at 20 µM was the most efficient concentration. Thus, we decided to chronically treat DED mice twice per day from day 7 until day 21 with one drop of M8-B at 20 µM or phosphate buffer saline (PBS). At day 21 post-surgery, to test corneal cold nociceptors responsiveness, we assessed electrical activities of the ciliary nerve fiber in ex vivo eye preparations using a multi-unit extracellular recording.

When the temperature of superfusion medium was set to 32–34 °C, the ongoing nerve impulse activities decreased by 23% in DED mice instilled with 20 µM M8-B compared to DED mice instilled with PBS (DED + PBS 86.78 ± 5.57 imp/sec vs. DED + M8-B (20 µM) 66.00 ± 11.43 imp/sec, *p* < 0.05; Figure 4A). When the temperature of the superfusion medium was decreased to 20 °C, the cold-evoked nerve activities reduced by 35.33% in DED mice chronically instilled with M8-B (20 µM) compared to DED mice instilled with PBS. (DED + PBS 127.71 ± 13.53 imp/sec vs. DED + M8-B (20 µM) 82.33 ± 4.27 imp/sec, *p* < 0.05; Figure 4B).

### 2.4. Chronic Topical Treatment with M8-B Reduced the Corneal Inflammation in the DED-Animals

At day 21 post-surgery, we assessed the effect of chronic M8-B (20 µM) topical treatment in DED mice on corneal layer alterations (i.e., superficial epithelium, sub-basal nerve plexus, and stroma) by in vivo confocal microscopy. Figure 5A shows that no difference was detected in the corneal superficial epithelium layer between PBS- and M8-B-treated DED mice. However, we noticed that M8-B topical chronic treatment diminished the number of long thin irregular dendritic cells in the sub-basal plexus (Figure 5B, red arrows) and refractive cells in the stroma compared with DED mice topically instilled with PBS (Figure 5C, red stars).

### 2.5. Chronic Topical Treatment with M8-B Reduced Inflammatory Gene Expressions in the TG of DED Animals

To investigate the impact of chronic M8-B (20 µM) topical treatment on peripheral sensitization mechanisms affecting the TG of DED mice, we assessed the expression of genes related to inflammation and neuropathic pain. Thus, the levels of tachykinin precursor 1 (*TAC1*) mRNA (Figure 6A) decreased significantly by 98% in M8-B-treated DED mice compared with PBS-treated DED mice (DED + PBS 3.56 ± 0.85 vs. DED + M8-B 0.05 ± 0.10, *p* < 0.01). The levels of interleukins-1β (IL1β) and interleukins 18 (IL-18) mRNA decreased by 29% and 68%, respectively, in M8-B-treated compared to PBS-treated groups (IL1β mRNA: DED + PBS 1.89 ± 0.21 vs. DED + M8-B 1.35 ± 0.05, *p* < 0.05; IL18 mRNA: DED + PBS 3.69 ± 0.92 vs. DED + M8-B 1.18 ± 0.07, *p* < 0.01; Figure 6A). We noticed that levels of the chemokine (C-C motif) ligand 2 CCL2 and CX3C chemokine receptor 1 (CX3CR1) mRNA decreased by 65% and 44%, respectively, in M8-B-treated vs. PBS-treated DED mice (CCL2 mRNA: DED + PBS 4.28 ± 0.96 vs. DED + M8-B 1.46 ± 0.09, *p* < 0.01; CX3CR1: DED + PBS 1.74 ± 0.36 vs. DED + M8-B 0.96 ± 0.97, *p* < 0.05; Figure 6A). Levels of prostaglandin E synthase 3 (PTGES3) and prostaglandin E receptor 3 (PTGER3) mRNA decreased by 62.80% and 64.79%, respectively, in M8-B-treated vs. PBS-treated DED mice (PTGES3: DED + PBS 3.98 ± 0.85 vs. DED + M8-B 1.48 ± 0.11, *p* < 0.05; PTGER3: DED + PBS 2.84 ± 0.72 vs. DED + M8-B 1.00 ± 0.21, *p* < 0.01; Figure 6A).

### 2.6. Chronic Topical Treatment with M8-B Decreased the Expression of Genes Implicated in Pain Conduction 

We investigated the impact of topical M8-B on genes implicated in pain conduction in the TG of DED mice treated with M8-B (20 µM) compared with DED mice treated with PBS.

#### 2.6.1. Intracellular Signal Transduction

The levels of mitogen-activated protein kinase (MAPK8) mRNA decreased by 50% in M8-B-treated DED mice compared with PBS-treated DED mice (MAPK8: DED + PBS 2.26 ± 0.58 vs. DED + M8-B 1.12 ± 0.05, *p* < 0.05; Figure 6B).

#### 2.6.2. Voltage-Gated Ion Channels

The levels of the sodium channel 1.8 (Nav1.8) and sodium channel 1.7 (Nav1.7) mRNA decreased by 39% and 23%, respectively, in M8-B-treated DED mice compared with PBS-treated DED mice (Nav1.8: DED + PBS 1.29 ± 0.27 vs. DED + M8-B 0.78 ± 0.03, *p* < 0.05; Nav1.7: DED + PBS 2.23 ± 0.70 vs. DED + M8-B 1.71 ± 0.01, *p* < 0.05; Figure 6). The levels of potassium channel 2 (KCNJ6) mRNA decreased by 70% (KCNJ6: DED + PBS 3.52 ± 1.91 vs. DED + M8-B 1.03 ± 0.06, *p* < 0.05; Figure 6C).

### 2.7. Pharmacological Blockage of TRPM8 Reduced Spontaneous and Mechanical-Evoked Ocular Pain Sensations in DED Mice

We previously reported that severe DED induced corneal mechanical allodynia and spontaneous pain behavior (decreased eye closing ratio) [41]. Here, we measured the mechanical sensitivity of the cornea in DED mice chronically instilled with M8-B (20 µM) or PBS. The corneal mechanical threshold increased by 200% in DED mice treated with 20 µM M8-B compared with DED mice treated with PBS (DED + PBS 0.01 ± 0.01 g vs. DED + M8-B (20 µM) 0.03 ± 0.01 g, *p* < 0.01; Figure 7A). We found that the spontaneous eye pain increased by 15% in DED mice treated with 20 µM of M8-B compared with DED mice treated with PBS (DED + PBS 0.7 ± 0.02 vs. DED + M8-B (20 µM) 0.8 ± 0.03, *p* < 0.05; Figure 7B). Moreover, we evaluated corneal epithelial damage using slit lamp examinations, after instillation a drop of 0.5% fluorescein, and observed superficial punctate keratitis (red arrows, Figure 7C) in the cornea of the DED mice chronically treated with M8-B (20 µM) or PBS. However, the fluorescein staining score was not significantly different between DED groups (DED + PBS 2.61 ± 0.22 vs. DED + M8-B (20 µM) 2.62 ± 1.16, *p* > 0.05; Figure 7C).

## 3. Discussion

DED patients suffer from abnormal hypersensitivity to normally harmless cold stimuli which is known as cold allodynia [42]. TRPM8 is a key channel for cold and pain perception [10,43,44,45]. Previous studies reported the effectiveness of M8-B, a selective and potent TRPM8 antagonist [46,47,48], on cold nociceptors via TRPM8 by decreasing deep body temperature in *Trpm8*^+/+^ mice and rats, but not in *Trpm8*^−/−^ mice, highlighting an on-target action [48]. Thus, in the present study, we explored whether M8-B could alleviate ocular pain associated with severe DED.

The relevant contribution of our study is the demonstration that chronic topical instillations of M8-B in mice suffering of severe DED blocks: (i) cold hyperresponsiveness of corneal nerves, (ii) corneal inflammation, and (iii) corneal mechanical allodynia and spontaneous ocular pain. So, our results confirmed the pivotal role of TRPM8 in acute pain and that the TRPM8 antagonist could be a promising treatment to alleviate chronic ocular pain caused by severe DED.

These results were obtained in adult male mice submitted to severe DED after the removal of two different functional (aqueous and lipid) glands which cause a drastic and permanent decrease of tear production by 97% [41]. Our DED model closely mimic both aqueous-deficient DED by excision of the extraorbital lacrimal gland (ELG) and evaporative DED by excision of the Harderian gland (HG), which produces an oily, lipid-enriched secretion [41]. The severe DED induced by this surgical procedure may be considered as a limitation since we cannot directly test the effectiveness of TRPM8 antagonist treatment on tear production. However, we developed a model of chronic DED associated with increased ocular nociception without the need to constantly inject chemical solutions with neurotropic activity (such as scopolamine) or to place mice in a desiccative stressful environment. Otherwise, a comparative study between adult female mice submitted to severe DED [41] with male DED and sham mice was previously performed on behavioral and electrophysiological responses related to the present study and no sex differences were noticed (data not shown).

Cool cells activity is increased by menthol, a specific TRPM8 agonist [18,19]. Here, we observed that DED mice developed a higher corneal menthol sensitivity (cold allodynia), which can be explained by the upregulation of TRPM8 mRNA expression observed in the ophthalmic branch of the TG. However, Hatta et al. showed that TRPM8 mRNA levels were not affected in corneal trigeminal neurons from a DED rat model obtained 10 days after extraorbital and infraorbital lacrimal glands removal [49]. In addition, TRPM8 protein levels in the cornea, conjunctiva, and TG were the same in sham animals and moderate DED rat model obtained 14 days after ELG excision [50]. This discrepancy of our results may be explained by (1) using a different DED model obtained by removing two functionally different glands (aqueous and lipid) vs. only aqueous glands. The excision of two functionally different glands at day 21 decreased tear production by 97% in our model [41] compared to 30% reported 10 days after ELG and infraorbital LG excision in rats [33] and 50% after 14 days ELG excision in rats [51], highlighting the severity of our DED model compared to moderate models of DED; (2) using post-surgery evaluation kinetics for the evaluation of TRPM8 expression at day 21 vs. day 10 or day 14 post ELG excision; and (3) monitoring TRPM8 mRNA expression in the ophthalmic branch of TG (responsible for both cornea and conjunctiva innervation) vs. only retrograde-labelled corneal trigeminal neurons. Altogether, our findings highlight an epigenetic regulation of the TRPM8 channel expression in the ophthalmic branch of TG during severe DED.

Ongoing ciliary nerve fiber electrical activity mostly results from cold nerve fiber activity, which represents half of the corneal sensory neurons [52,53]. Thus, the increase in ongoing ciliary nerve fiber activity in our severe mice model of DED may result from the triggering of corneal cold nociceptors, as was also reported in DED guinea pigs obtained after ELG removal [32]. Corneal cold-thermoreceptor fibers are classified into two groups: high-background low-threshold cold thermoreceptors (HB-LTCs) have a background firing rate at the normal temperature of the cornea (~34 °C), which increases with small temperature decreases (<1 °C); and low-background high-threshold cold thermoreceptors (LB-HTCs), which need large temperature reductions to increase their firing rate [15,54]. HB-LTCs control basal tearing and blinking rates, whereas LB-HTCs evoke sensations associated with corneal dryness [55]. Whereas acute topical treatment with M8-B (20 and 30 µM) decreased only the spontaneous ciliary nerve fiber activity at 32 °C, we observed that 14 days’ treatment by 20 µM M8-B significantly decreased the spontaneous and cold-evoked ciliary nerve fiber activity in DED mice. These observations confirm the effectiveness of M8-B on cold responsiveness by acting on both HB-LTCs and LB-HTCs, which may lead to modulating blinking rate and corneal dryness in DED mice. Our anatomical and electrophysiological results unveiled TRPM8-dependent changes in corneal neurons during severe DED, suggesting that increased ongoing and evoked nerve activities in DED animals are modulated via TRPM8 channels.

Ocular surface alterations and inflammation are the core mechanism of DED [56]. We previously found corneal superficial punctate keratitis and an increase in inflammatory cells in the sub-basal plexus and corneal stroma in DED animals [41]. In this study, we demonstrated that chronic pharmacological blockade of TRPM8 by chronic instillations of M8-B considerably decreased corneal inflammation in DED animals. However, this topical treatment was not effective in reducing severe ocular surface epitheliopathy observed in DED mice. This can be explained by the radical reduction in tear secretion (by 97%) and the lack of lubrication causing chronic mechanical alterations on the corneal epithelial integrity.

TRPM8 has been implicated in inflammation in a model of asthmatic mice where cold stimulus promoted inflammation via TRPM8 [57,58]. Thus, based on our observations and literature findings, we suggest a link between TRPM8 and corneal inflammation associated with DED. The anti-inflammatory effect of TRPM8 blockade was not limited to the cornea; it also led to changes in the ipsilateral TG. A drastic decrease in the immune receptors and inflammatory mediator’s mRNA levels of IL-1ß, IL-18, CCL2, prostaglandins, and CX3CR1 was observed in the TG of DED animals chronically instilled with M8-B. These receptors and inflammatory mediators are known to be increased in DED and have been implicated in the spread of inflammation from cornea to the TG [59,60,61]. Thus, our results clearly suggest that M8-B can decrease the spread of inflammation from the cornea to TG in DED animals via inhibition of TRPM8. 

Sodium and potassium currents have been associated with cold responses [62,63,64]. In the context of DED, Kovacs et al. reported that ocular cold nociceptor activities are mediated by enhanced Na^+^ currents combined with diminished K^+^ currents in trigeminal corneal cold sensory neurons in a DED guinea pig model produced after ELG removal [32]. Our findings showed decreases in Nav1.8, Nav1.7, and KCJN6 channel mRNA levels in the TG from DED mice treated with M8-B. Since Na^+^ and K^+^ channels play a role in ocular cold responsiveness, our result showing that TRPM8 antagonist decreases the gene expressions of Na^+^ and K^+^ channels in the TG highlights the effectiveness of the TRPM8 antagonist in blocking the exacerbated cold responses of the cornea observed in DED-mice.

Substance P (SP) was found to be required for corneal cold nociception [65]. SP is encoded by the *Tac1* gene and is known to modulate neurogenic inflammation [66,67]. Here, we report that M8-B decreased the *Tac1* mRNA level in the TG of DED animals, underlining a reduction in neurogenic inflammation and the excitation of peripheral free terminals in DED animals. This hypothesis is supported by the decrease in intracellular signaling pathway MAPK8 mRNA levels observed in the TG of DED animals treated with M8-B, emphasizing a lesser propagation of nociceptive messages. 

Although much attention has been paid to the role of TRPM8 in cold allodynia and hyperalgesia, some studies have explored the role of TRPM8 in mechanical allodynia. Klein et al. reported that topical application of menthol induced mechanical allodynia in wild type (WT) rats [68], whereas TRPM8 inhibitors were shown to reduce cold and mechanical allodynia in acute and chronic pain models [29,30]. In line with these studies, we observed that chronic M8-B instillations decreased the mechanical corneal hypersensitivity and spontaneous eye pain developed in DED animals. One explanation for the anti-mechanical allodynic effect of TRMP8 antagonist is possible crosstalk between TRPM8 and a polymodal channel (TRPV1) that can be activated by mechanical stimulations. In agreement with our hypothesis, Hatta et al. reported increases in TRPM8 and TRPV1 co-expressions in the TG neurons of DED rats obtained after extraorbital and infraorbital lacrimal glands removal [49]. A recent study demonstrated an overexpression of TRPV1 in TRPM8+ cold-sensing fibers, which caused cold allodynia in a mouse model of DED obtained by ELG excision [65]. For the first time, this finding showed that TRPV1 activity is required for corneal cold nociception. Altogether, we suggest that TRPM8 blockage influences TRPV1 activity and leads to a decrease in corneal mechanical hypersensitivity.

## 4. Methods

### 4.1. Experimental Animals

Seven- to eight-week-old adult male C57BL/6 mice (average weight 23.48 ± 0.04 g; Janvier Labs, Le Genest Saint Isle, France) were maintained under controlled conditions (22 ± 1 °C, 60% ± 10% relative humidity, 12/12 h (hour) light/dark cycle, food and water ad libitum). All animal procedures were performed in strict accordance with institutional guidelines for the care and use of experimental animals approved by the European Communities Council Directive 2010/63/UE (APAFIS #1501 2015081815454885 v2, 30 September 2015).

### 4.2. Surgical Procedures

Unilateral (right side) ELG and HG excision was performed under ketamine (80 mg/kg intraperitoneal (i.p.) and xylazine (8 mg/kg i.p.) anesthesia, as recently described [41]. Before surgery, a drop of lacrimal gel (Lubrithal™, Hamilton, Canada) was applied to both eyes. Under an operative microscope (Leica-Alcon II, Wetzlar, Germany), an 8 mm skin incision was made on the temporal side to expose and remove the ELG. After dissociating the conjunctival tissue above the orbital cavity near the internal canthus, the HG was carefully removed. Complete removal was verified by inspecting the surgical area for any remaining glandular tissue. The skin incision was then sutured using 6.0 braided silk sutures (6-0 Vicryl, Ethicon, Dulmen, Germany). A drop of iodine solution was applied to the incision to avoid bacterial infection. For sham mice, an incision was made in the same zone without touching the glands. The mice were placed in warm (30 °C) cages to recover from the surgery.

### 4.3. Drugs

M8-B hydrochlorid: *N*-(2-aminoethyl)-*N*-(4-(benzyloxy)-3-methoxybenzyl)thiophene-2-carboxamide hydrochloride, *N*-(2-Aminoethyl)-*N*-[[3-methoxy-4-phenylmethoxy) phenyl] methyl]-2-thiophenecarboxamide hydrochloride) (Scheme 1) (SML0893-5MG, Sigma-Aldrich, St. Louis, MO, USA) was dissolved in sterile distilled water at the concentration of 2 mg/mL and then diluted in phosphate-buffered saline (PBS) 1× to obtain 10, 20, and 30 µM solutions.

Batch Molecular Formula: C22H24N2O3S·HCl. Batch Molecular Weight: 432.96. HPLC: Shows ≥98% purity. M8-B is a selective and potent antagonist of the TRPM8 channel. M8-B blocked cold-induced and icilin or menthol-induced activation of rat, human, and murine TRPM8 channels with the half maximal IC50 values ranging from 7.8 to 64.3 nM, and it did not block other TRP channels (IC50 values > 20 mM).

Menthol (Sigma-Aldrich, St. Louis, Missouri, United States) was diluted in pure castor oil (Sigma-Aldrich, St. Louis, Missouri, United States) to obtain a 50 µM solution.

### 4.4. Experimental Study

Twice per day, DED mice received ocular instillations of M8-B (20 µM) or PBS from days 7 until 21. For topical ocular administration, the cage order was randomized daily, and the experimenter was blinded to the treatment group. The same experimenter performed all experiments, administrated the study treatment, and assessed the effect. The experimenter was blinded to the treatment group.

### 4.5. Behavioral Tests

For behavioral tests, mice were placed in the testing room (at least 30 min before the start of the experiments) and the order of testing was randomized. The immobilization (handling-restraint) lasted a maximum of 20 s to reduce any potential stress and anxiety. Behavioral experiments were conducted at day 21.

### 4.6. Measurement of Corneal Sensitivity to Mechanical and Cold Stimulation

Mechanical corneal sensitivity was monitored using von Frey filaments, as previously described [41]. Various forces of calibrated von Frey filaments (0.008 to 0.04 g) were applied to the center of the cornea of manually immobilized mice. The mechanical threshold corresponded to the eye-blink response. For corneal cold sensitivity, 10 µL of a 50 µM menthol solution was applied to the right eye. Animals were immediately placed in individual cages and the eye closing ratio was measured 5 min after menthol ocular application.

### 4.7. Spontaneous Pain: Eye Closing Ratio Measurement

Spontaneous eye closure is an accurate index used for monitoring spontaneous eye pain [41] and is one of the quantitative measures of the grimace scale, which is used to monitor spontaneous pain behavior [69,70]. The eye closing ratio was calculated based on photographs of awake and unconstrained mice and corresponds to the height/width ratio of palpebral closure. Width is the distance between the internal and external canthus and height is the distance between the edge of the upper and lower eyelids, going through the center of the cornea. Images were captured by a digital camera using EyeSuite™ software (Bern, Switzerland).

### 4.8. Fluorescein Staining Score

The corneas of lightly anesthetized mice were washed with sterile 0.9% NaCl. Then a drop of fluorescein (Fluorescein Faure 0.5%) was placed into the conjunctival sac of the right eye. Ocular surface staining was then evaluated using a slit lamp under a cobalt blue light (peak of approximately 400 nm). Images were captured by a digital camera using EyeSuite™ software (Koeniz, Switzerland). The extent of corneal damage (width and intensity of fluorescein take-up area) was scored according to the following scale: 0, absence of staining; 0.5, slight punctate staining; 1, diffuse punctate staining; 2, diffuse staining covering less than one third of the cornea; 3, diffuse staining covering more than one third of the cornea; and 4, staining covering more than two thirds of the cornea [41,71].

### 4.9. In Vivo Confocal Microscopy

An in vivo laser confocal microscope (IVCM, Heidelberg Retina Tomography (HRT) II/Rostock Cornea Module (RCM); Heidelberg Engineering GmbH, Heidelberg, Germany)) was used to examine the entire cornea of anesthetized mice [72]. The images covered a 400 × 400 μm area with a transversal optical resolution of 2 μm and an axial optical resolution of 4 μm (Heidelberg Engineering, Heidelberg, Germany). Approximately 100 images were acquired per animal. Image acquisitions always started in the center of the cornea at the level of the superficial epithelium. Peripheral acquisitions were then performed similarly.

### 4.10. Multi-Unit Extracellular Recording of Evoked Ciliary Nerve Fiber Activity in Ex Vivo Eye Preparations

Spontaneous ciliary nerve fiber activity was determined at day 21 as previously reported [41,72]. The spontaneous ciliary nerve activity’s recordings were performed by superfusing the cornea with superfusion saline solution at 33 ± 1 °C and pH 7. Briefly, mice were euthanized and the eye was placed in a two-compartment chamber. The cornea was continuously superfused at a rate of 3 mL/min at 33 ± 1 °C with a superfused saline solution (133.4 mM NaCl, 4.7 mM KCl, 2 mM CaCl_2_, 1.2 mM MgCl_2_, 16.3 mM NaHCO_3_, 1.3 mM NaH_2_PO_4_, and 7.8 mM glucose) saturated with O_2_ and adjusted to pH 7.4 by bubbling with 95% O_2_ and 5% CO_2_. Multi-unit extracellular electrical activity of the ciliary nerve was recorded using a suction electrode (Ag/AgCl). The signal was filtered (300–5000 Hz), amplified (10,000×) (A-M Systems, Sequim, WA, USA), and digitalized by Spike 2 data analysis (CED Micro1401, Cambridge Electronic Design, Cambridge CB24 6AZ, United Kingdom) at a sampling frequency of 10,000 Hz. The cornea was superfused with superfusion saline solution for 30 min to stabilize the preparation before performing electrophysiological recordings. Extracellular spontaneous ciliary nerve fiber activity is defined as impulses per second (imp/s).

The ciliary nerve evoked activity’s recordings were performed by superfusing the cornea with superfusion saline solution at 20 ± 1 °C and pH 7. A thermistor sensor (included in the CL-100 Bipolar Temperature Controller, Warner Instruments, CT, USA) monitored the temperature at the exit of the corneal superfusion. The acute application of M8-B was evaluated by superfusion of the cornea with 10, 20, and 30 µM M8-B diluted in the superfusion saline solution.

### 4.11. RT-qPCR Analysis

#### 4.11.1. Tissue Preparation for RT-qPCR Analysis

Twenty-one days after surgery, the animals were deeply anesthetized with a 300 μL mixture of ketamine (80 mg/kg) and xylazine (8 mg/kg) and transcardially perfused with 10 mL 0.9% NaCl solution. Ipsilateral TGs were carefully removed and stored at −80 °C.

#### 4.11.2. RT-qPCR Analysis

RNA extraction from the ipsilateral TG was performed with a NucleoSpin RNA Purification II Kit (NucleoSpin RNA S, Duren, Deutschland, Germany). RNA quality and concentration were then measured by the NanoDrop method (Thermo Scientific, Waltham, MA, USA). Then, reverse transcription was performed by iScript cDNA Synthesis Kit (Bio-Rad, Hercules, CA, USA) according to the manufacturer’s instructions. PCR was performed with 300 ng cDNA for each simple. RT-qPCR was performed with SsoAdvanced Universal SYBR^®^ Green Supermix (Bio-Rad, Hercules, CA, USA), and a pain, neuropathic, and inflammatory (SAB Target List) M96-well plate (Bio-Rad, Hercules, CA, USA; ref: 10034393). The *GAPDH* gene was used as the endogenous reference for each reaction; mRNA levels were calculated after normalizing the results for each sample with those for *GAPDH* mRNA. The 2^-ΔΔCt^ method was used to analyze the relative differences in specific mRNA levels between groups.

### 4.12. Tissue Preparation for Fluorescent In Situ Hybridization

Twenty-one days after surgery, anesthetized mice were transcardially perfused with 10 mL 0.9% NaCl solution followed by 40 mL 4% (*w*/*v*) paraformaldehyde in PBS. Next, TGs were immersed in 10%, 20%, and 30% sucrose in PBS and then conserved in isopentane with liquid nitrogen and stored at −80 °C. TGs (12 μm) were cut on a cryostat (Leica CM 3050 S, Wetzlar, Germany) and mounted on Superfrost slides (Thermofisher scientific, Waltham, MA, USA).

### 4.13. Fluorescent RNAscope In Situ Hybridization

Fluorescent in situ hybridization studies was performed according to the protocol for fixed frozen tissue using the RNAscope Fluorescent Multiplex Reagent Kit v2 assay (Advanced Cell Diagnostics, Newark, CA, USA). Tissues were washed with PBS and treated with hydrogen peroxide (RNAscope, Bio-techne, Minneapolis, Minnesota, United States, ref# 322335) for 10 min at RT and washed in autoclaved distilled water. Using a steamer, tissues were treated with distilled H_2_O for 10 s at 99 °C and then moved to RNAscope 1× target Retrieval Reagent (RNAscope, Bio-techne, Minneapolis, Minnesota, USA, ref# 322000) for 5 min at 99 °C. Tissues were washed with autoclaved distilled water and transferred to 100% alcohol for 3 min. After, tissues were treated with RNAscope Protease III (RNAscope, Bio-techne, Minneapolis, MA, United States, ref#322337) for 30 min at 40 °C and washed with autoclaved distilled water. Species-specific target probe TRPM8-C3 (420451 C3) was used. Sections were treated with the probe; negative (ref# 320871) and positive (ref# 320881) controls and were hybridized for 2 h at 40 °C in a humidified oven (RNAscope, Bio-techne, Minneapolis, MA, USA, HybEZ oven with HybEZ humidity control tray). A series of incubations were then performed to amplify the hybridized probe signals and label target probes with the assigned fluorescence detection channel (target probe was labelled with the assigned fluorescence detection channel (PerkinElmer, Waltham, MA, USA. Opal 650 FP1496A). Nuclei were stained using a DAPI nuclear stain (RNAscope, Bio-techne, Minneapolis, MA, USA, Ref# 323108) for 30 s at RT. Slides were mounted with ProLong Gold Antifade Mountant (Thermofisher scientific, Waltham, MA, USA, Ref# P36934) onto glass sides and cover slipped.

### 4.14. Microscopic and Nanozoomer Analysis

TGs were examined using a nanozoomer. The area within the field of interest covered by the TRPM8 mRNA profiles relative to the total area of the measured field was analyzed to detect mRNA levels in the ophthalmic branch in the TG. The same gray threshold level was applied to all sections of the same series. TIFF images were analyzed using NIH ImageJ software.

### 4.15. Statistical Analyses

The data obtained in different groups were compared using the appropriate paired parametric or nonparametric statistical test, as indicated. For statistical analysis, the Kolmogorov–Smirnov test was performed, followed by the parametric *t*-test or nonparametric Mann–Whitney or parametric one-way ANOVA test (GraphPad Software, La Jolla, CA, USA). All *p*-values were considered statistically significant when values were <0.05. All results are presented as the mean ± standard error of the mean (SEM).

## 5. Conclusions

Overall, our study showed that pharmacological blockade of corneal-expressed TRPM8 (by chronic instillations of M8-B) significantly reduced corneal neurosensory abnormalities in a preclinical model of DED, thus reinforcing the pharmacological concept that TRPM8 antagonists may be of use as a local antalgic to relieve ocular pain in cases of severe DED.

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
