# Peer review of "TRPM8: A Therapeutic Target for Neuroinflammatory Symptoms Induced by Severe Dry Eye Disease"

_ijms, 2020, doi:10.3390/ijms21228756_

Round 1

Reviewer 1 Report

Review

The manuscript “TRPM8: A therapeutic target for neuroinflammatory symptoms induced by severe dry eye disease.” By Darine Fakih and co-workers explored the effect of TRPM8 blockade for the treatment of ocular pain in severe dry eye. The authors say that an antagonist of TRPM8 is working against the DED animal model. The data are straightforward and easy to follow. Some critical technical details should be addressed to improve the manuscript.

The chemical structure of the antagonist is not given, this is a requirement

The properties of the antagonist are not given, e.g. the IC50 against various agonists, e.g. menthol, icilin, etc. If you use an antagonist to show receptor specificity, then the IC50 value must be given in molar units, the agonists being antagonized must be identified, the chemical structure of the antagonist must be shown, as well as the methods of synthesis and the purity of the tested agent, and how it is prepared, formulated and tested. Data on selectivity must be included.

Validation of the DED model and the baseline degree of DED parameters (e.g corneal epithelial damage, tear secretion) as well as its follow up data after the treatment must be presented in the data.

Reviewer 2 Report

Dry eye disease (DED) is associated with ocular surface inflammation and corneal pain syndrome. In this study, the author evaluated the effectiveness of repeated instillations of transient receptor potential melastatin 8 (TRPM8) ion channel antagonist (M8-B) on a mouse model of severe DED, which was induced by the excision of extra-orbital lacrimal and Harderian glands. This study provides new insight into the effectiveness of the TRPM8 blockade for alleviating corneal pain syndrome associated with severe DED. Accordingly, the author implied an insightful treatment for ocular pain management in severe DED patients.

However, the author did not follow the format of this journal and did not provide a figure in the manuscript. Therefore, the author should correct the article format and provide the figures mentioned in the text.

Round 2

Reviewer 1 Report

How can the authors verify they have completely successfully removed the lacrimal gland in every of their model? Tear volume must be measured along with their TRPM8 antagonist experiment to confirm persistent tear volume reduction. Since the rodents have extra- and intraorbital lacrimal gland (also Hardarian), the reduction of the tear volume varies depending on the contribution of the extra- and intraorbital lacrimal gland. The reproducibility is the key issue in the lacrimal gland excision model. Please refer to Dr.Kinoshita’s paper for more information (A new dry eye mouse model produced by exorbital and intraorbital lacrimal gland excision. 2018 Scientific Report). The authors should perform additional experiments to strengthen their data rather than simply adding their excuses in the discussion.

Also, DED is classified as aqueous-deficient dry eye (ADDE) and evaporative dry eye (EDE). The model that the author’s employed explains only ADDE by removing the lacrimal unit. This must be discussed.

Reviewer 2 Report

The work is good. I do recommend the manuscript for publication.